# Study on the Measurement and Influencing Factors of Care Service Demand of Disabled Elderly in Urban and Rural China

**DOI:** 10.3390/ijerph191711112

**Published:** 2022-09-05

**Authors:** Haixia Jiang, Suhua Xiao, Hongwei Hu, Haotian He

**Affiliations:** 1School of Public Administration, Hunan University, Changsha 410082, China; 2School of Public Administration and Policy, Renmin University of China, Beijing 100872, China

**Keywords:** population aging, urban and rural disability elderly, disadvantaged families, care service demand, MIMIC structural equation model

## Abstract

Caregiving services are currently the weakest part of China’s social security system for the elderly. It is well needed to investigate the fac-tors affecting the unmet care needs of the elderly with disabilities. Based on the Behavioral Model of Health Services Use (BMHSU), this paper explores the needs and influencing factors of care services for the disabled elderly in urban and rural areas using data from the 2018 Project of Social Policy Support System for Disadvantaged Families in China. The demand for care services of the disabled elderly in central and western areas is significantly higher than that in eastern, along with that in rural areas significantly higher than that in urban areas. The demands for care services of the disabled elderly in urban and rural areas are significantly affected by tendency factors, resource factors, and need factors. Urban and rural attributes, worried pension and LCI are the common influencing factors for the care service demand of the disabled elderly from economically disadvantaged families and ordinary families. The demands for care services of the disabled elderly were associated with tendencies, resources, and needs, increased chronic disease prevention and mental health services benefit caregivers.

## 1. Introduction

As of the end of 2021, there were 267 million people aged 60 or above, accounting for 18.90% of the total population, and 201 million people aged 65 or above, accounting for 14.20% of the total population, according to China’s National Bureau of Statistics [1]. Compared with the results of China’s sixth population census, the proportion of the population aged 60 and over and the proportion of the population aged 65 and over increased by 5.64% and 5.33% respectively in the past decade. According to the fourth sample survey on the living conditions of the elderly in urban and rural China, by 2015, the disabled and semi-disabled elderly accounted for 18.30% of the total elderly population, or 40.63 million people [2]. The number of disabled aged people in China reached 52.71 million in 2020, and is expected to exceed 77.6568 million in 2030. If no preventive and control measures are taken, the number of disabled aged people will further increase to 95.368 million in 2050 [3]. The increasing risk of disability in the aged population is an unavoidable problem.

With the increasingly serious problems of population aging, advanced aging, and disability, the scale of disabled, and semi-disabled elderly is rapidly expanding. How to properly meet the demands of disabled elderly care services has become a common problem for tens of millions of Chinese families. On the one hand, with the deepening of industrialization and urbanization, the phenomenon of population mobility is becoming more and more frequent. Affected by the improvement of housing conditions and the independent living of young people after marriage, the miniaturization of family structure, the separation of young children from the elderly, and the empty nesting of the elderly population are becoming increasingly common [4], which has a certain impact on the traditional family care function. On the other hand, with the advancement of women‘s social status, the rapid change in science and technology, and the increasingly fierce competition, family members are facing more pressure in life and work. The rise of family burden coefficient leads to family members’ inability to care for the elderly [5]. The care and nursing of the elderly has become a real problem for many families. It can be seen that the major changes in social structure and family structure have shaken the equilibrium state of care services to a certain extent [6], resulting in the phenomenon of uncoordinated and mismatched population age structure and economic and social development [7]. The proportion of disabled elderly who rely on communities and institutions to provide care services will also continue to increase in the future, and the care demands of the elderly deserve further attention.

In the early 21st century, the World Health Organization and the World Bank began to actively advocate for a basic consensus around the world on issues related to long-term care, clarifying the basic concepts of long-term care from the perspective of caregivers or caregivers, respectively. In 2000, in the WHO’s International Consensus on Building Longterm Care Policies for Older Persons, it was pointed out that long-term care service is a nursing care activity carried out by informal care providers such as families, friends or neighbors, and formal care providers such as health and social professionals, as well as volunteers. The purpose is to ensure that individuals lacking self-care ability can maintain the highest level of quality of life and enjoy the greatest possible independence, autonomy, participation, personal satisfaction, and human dignity according to their choices and needs [8]. Chinese scholars Tang & Feng also clarify the meaning of long-term care, “care” refers to life care for the existence of care-dependent disabled people; “nurse” is the nursing and rehabilitation services that are difficult to distinguish between life care and nursing and rehabilitation [9]. In summary, care services as a comprehensive concept encompass both the care needs of individual physical illnesses and the care needs resulting from injuries to mental and social adaptation caused by health.

In order to actively respond to the aging population, meet the long-term care demands of disabled people, protect the basic rights and interests of disabled people, and improve the decent and dignified quality of life of disabled people, the Chinese government has made it clear that local governments should speed up the establishment of an evaluation mechanism for elderly care services, establish and improve a subsidy system for the elderly with financial difficulties and disability, and provide institutional guarantee for elderly care services for the elderly with financial difficulties and disability [10]. Since 2016, China has carried out two trials of the long-term care insurance (LCI) system in 29 cities and regions. After six years of practical exploration, the trial areas have basically formed a policy framework and operation mode for a long-term care insurance system that is compatible with regional development. Now, the LCI has covered 134 million people in 49 pilot cities across the country, with a total of 1.52 million people receiving benefits [11], laying a foundation for the development of the long-term care system.

However, under the background of social transformation and urban-rural imbalance, care services are the weakest part of the current social security system for the elderly [12]. This is especially true for the poor elderly living on subsistence allowances and marginalized families in urban and rural areas, who face the pressure of inadequate basic pension security and weak family support. Meanwhile, the coverage of long-term care insurance in the trial phase only covers the insured population with basic medical insurance for urban employees, and mainly solves the basic nursing support demands of severely disabled people. Many urban and rural elderly are temporarily unable to obtain benefits through participating in LCI due to the “limited” coverage of the insurance. The relevant departments failed to integrate care service subsidies, old age subsidies, elderly disabled subsidies, and long-term care insurance [13], and the institutional advantages of long-term care insurance cannot be effectively played. The “mismatch” pattern of medical and nursing resources can no longer meet the current demands of care services for the disabled elderly in urban and rural areas. Therefore, in the process of promoting rural revitalization and the construction of new urbanization, more attention should be paid to the care services for the disabled elderly in urban and rural areas, adhering to the demands of the elderly, ensuring the rights and interests of the elderly, and accelerating the improvement of the care service system for the disabled elderly.

As an important part of the long-term care system, care service will affect the pilot promotion and policy implementation of the LCI system. In addition to “limited coverage” and “misallocation of resources”, what are the influencing factors behind the unmet care demands of the disabled elderly? This is the purpose and focus of this paper. The pilot of China’s long-term care insurance system is expanding, and the long-term care service system is still under construction. The level of demand for care services for urban and rural disabled elderly and its influencing factors are worthy of further exploration.

With the deepening of population aging and the increasing risk of disability, it is particularly critical to pay attention to and solve the demand for care services for urban and rural disabled elderly. Existing research on the needs of disabled elderly care services measurement indicators are relatively single, more limited to the study of a province or region. In general, there is a lack of systematic theoretical analysis framework, and the understanding of the influencing factors that affect the demand for care services of the elderly is not deep enough [14,15,16,17,18,19]. It fails to fully reflect the demand for care services for the elderly, and lacks the objectivity of research. Consequently, this study focuses on the demands of the disabled elderly in urban and rural areas for medical care, nursing, and rehabilitation services, combined with the BMHSU and the survey data of the social policy support system construction project for disadvantaged families in China in 2018, and the needs and influencing factors of care services for the disabled elderly in urban and rural China are explored using MIMIC structural equation model.

## 2. Literature Review and Theory

### 2.1. Literature Review

#### 2.1.1. Care Service Demand and Measurement

As an integrated service arrangement for the provision of living care and medical care to the elderly who have partially or completely lost their ability to take care of themselves, care services include basic daily care, professional medical care, nursing and rehabilitation services, spiritual comfort, and social support [20,21,22,23]. In the WHO’s International Consensus on Building Long-term Care Policies for Older Persons and the Global Report on Ageing and Health, long-term care is a nursing care activity carried out by formal and informal care providers, as well as volunteers, to ensure that individuals with persistent inherent disability or risk of corresponding disability receive financial, social and legal support, and to maintain a level of physical functioning that enables them to acquire fundamental rights, freedoms and human dignity [8,24], through the provision of services such as emergency medical and mental health care. The care services demands for the elderly include medical treatment, rehabilitation, and nursing, etc.

Existing studies on care services demand are mainly results-oriented, and are generally measured by asking interviewees whether they have the demand for care services or care insurance through questionnaires [14,15,16,17,18,19]. Some scholars measure the demand for care services by selecting single or multiple proxy variables such as “the elderly in nursing homes” and “the elderly dependent on long-term care”, considering that the current care insurance in China is still in the pilot stage and there are limitations in the coverage of care services [25,26,27,28]. Only a few scholars construct latent variables of care service demand from aspects of life care, medical service, and nursing service from the perspective of multi-dimensional explicit variables, and comprehensively measure and analyze the demand for care service [29,30,31].

#### 2.1.2. Care service Demand Satisfaction

Existing studies have shown that due to the relatively poor economic status of the elderly, the lack of ability to resist various risks, and especially the vulnerability of the elderly in urban and rural disadvantaged families in terms of social status and economic ability, the care service needs of the elderly from urban and rural disadvantaged families have not been effectively satisfied [32,33,34]. Gu & Vlosky used CLHLS2005 survey data to systematically assess the long-term care needs of China’s elderly population. The results show that nearly 3.5 million Chinese elderly need long-term care services, but are unable to obtain long-term care services, and the proportion is close to 60%. It is predicted that the number of elderly with unmet care services could increase to 16 million by 2050 [27]. Elderly living in rural areas have a higher level of unmet care demands than older people in urban areas [31,35]. On the other hand, even if urban communities can provide care services such as personal care, medical treatment, and psychological counseling, there are still 51.3~55.5% of the elderly whose needs for community care services cannot be met [36]. Under the predicament of limited care resources and uneven regional distribution, urban and rural residents are also limited by their eligibility for access to care services, which cannot ensure sufficient financial or human resources to meet the rehabilitation care demands of the elderly [37].

#### 2.1.3. Factors Influencing the Care Services Demand

The demand for care services of the disabled elderly in urban and rural areas is inseparable from the comprehensive effects of individual, family, and social factors Even if the elderly have a tendency to use health services, they are still affected by factors such as the policy environment, personal and family capabilities, and health conditions [38]. First, whether the disabled elderly choose care services is related to their own behavioral preferences and value judgments. Factors such as age, gender, region, and education level have a significant impact on the demand for elderly care services [14,39]. However, Zhu & Guo pointed out that gender, marriage, and urban-rural attributes are not factors that lead to disability and nursing needs, and the occurrence of disability risk and nursing needs are only related to age and education level [25].

Besides, the choice of care services for the disabled elderly in urban and rural areas is not only determined by individual preference, but also by the ability of the elderly to obtain care services and the availability of care service resources. The better economy, the more children, availability of family care, the less likely the elderly will need care services, while the elderly participating in basic pension insurance or medical insurance will lead to the need for elderly care services [31,40]. Hu, Si & Li also pointed out that the economic status of individuals significantly affects the demand for care services, and the socioeconomically disadvantaged groups are more likely to need care services [41]. However, Liao found that the number of children in the family and whether they participate in the new rural cooperative medical system, or the new rural insurance have no significant impact on the demand for long-term care of the disabled elderly in rural areas, and the demand for care services is not affected by social security resources [19].

In addition, Nieboer, A. et al. found that different elderly groups have different values for long-term care services, and the value of care services depends to some extent on the social background, including physical, spiritual, and social vulnerability [42]. Therefore, the health status of the elderly will also affect the care needs of the elderly, and chronic diseases are significantly related to the care needs of the elderly [15]. Physical, mental, or social health are important factors influencing the need for care services for the elderly [41,43,44,45], and the elderly with severe disabilities and chronic diseases need not only daily care, but also medical care and rehabilitation training care services [44].

Under the background of the lack of coordination in the allocation of care resources and the heterogeneity of individual characteristics of the elderly in China, the needs of elderly care services have not been met, and more attention should be paid to research on the care service needs of the disabled elderly and its influencing factors. Meanwhile, the current research on the measurement of elderly care service needs still lacks a comprehensive measurement perspective and more scientific measurement methods. This study primarily explores and analyzes the influencing factors of the demand for care services for the disabled elderly in urban and rural areas from three aspects: tendency factor, resource factor, and need factor, based on the BMHSU and the data from 2018PSPSSDFC, using the MIMIC model. Daily care, medical services, health education, rehabilitation nursing, psychological comfort, and social support are selected as observable variables, and the latent variables of the needs of urban and rural disabled elderly care services are constructed from the dimensions of multiple care services.

### 2.2. Theory and Hypotheses

The Behavioral Model of Health Services Use (BMHSU) originated from the behavioral model of family health service demand first proposed by American medical sociologist Dr. Anderson in 1968, and has been widely used to systematically analyze the influencing factors of medical and health utilization behavior. Peng et al. used the CLHLS2014 survey data and took the BMHSU as a theoretical framework to analyze the influencing factors of the use of long-term care services among the disabled elderly in China from forward leaning factors, enabling factors, and demand factors [38]. Sun et al. constructed an “analytical framework for pension decision-making behavior” based on BMHSU, and analyzed the influencing factors of rural elderly’s pension decision-making behavior [46]. Zeng et al. used the CHARLS2015 tracking survey data, and based on the BMHSU framework, to analyze the influencing factors of the elderly’s medical treatment behavior from the aspects of tendency characteristics, enabling resources, and medical needs [47]. Chen & Wang analyzed the unmet care needs and influencing factors of the disabled elderly living alone by introducing community factors based on the BMHSU [48].

Based on the above-mentioned scholars introducing the BMHSU as the theoretical analysis framework of the research, this article believes that the BMHSU can grasp the main characteristics of the groups receiving care service from a more comprehensive perspective, and the model can incorporate the multi-layered factors that affect the demand for care services into a relatively mature and stereotyped analysis framework to avoid the random selection of influencing factors.

The theoretical analysis framework consists of four parts in this article, as shown in Figure 1. (1) External environment, that is, the policy environment of the long-term care service system, facing the objective needs of the increasing number of people with disabilities and dementia for long-term care security. (2) Subject characteristics. The subject characteristics of the disabled elderly in urban and rural areas, as the basic component of the Anderson model, reflect the impact of the “individual level” subjective conditions of the disabled elderly in urban and rural areas on their care service demand behaviors. The article mainly explores and analyzes the following three aspects: (a) Tendency factors are the individual preference characteristics of disabled elderly choosing to use care services, which mainly cover three indicators: demographic characteristics, social structure characteristics, and attitudes and values. They respectively represent the possibility of the elderly needing care service, biological characteristics; the social status of the elderly mainly includes educational level, urban and rural attributes, and other sociological indicators; the cognitive attitude of the elderly mainly involves their basic attitude towards future pension concerns. (b) Resource factors are the ability of the disabled elderly to obtain care services and the availability of care service resources, which are indirect factors in the demand for care services, mainly including the elderly economic conditions, daily access to loved ones or family care, participation in LCI and other personal, family, and social care services resources. (c) Need factor is the perceived need for care services generated by the disabled elderly based on the subjective judgment of their own health and disease status and the care service evaluation needs to be generated by professional measurement and evaluation of the health of the elderly, which is the precondition and direct influence factor for the occurrence of care service demand behavior. (3) Service process, that is, the behavioral process of using care services for the disabled elderly in urban and rural areas, mainly includes the process of self-regulation and the process of service utilization: “Self-regulation” refers to the process by which the elderly individuals improve their health status by creating conditions to improve their living habits, enhance physical exercise and other self-care methods; “Service process” refers to the process in which the elderly obtain basic life care, medical care, rehabilitation, and otherwise. (4) Service results, that is, the nursing service effect feedback and service quality evaluation obtained by the re-evaluation of their own health status after using the nursing service for the disabled elderly in urban and rural areas.

This study was preliminarily designed to explore the measurement and influencing factors of care services demands of disabled elderly in urban and rural areas in China. The hypotheses included the following:

**Hypothesis** **1.**
*In terms of tendency factors, disabled elderly who are older, live in rural areas, have relatively low levels of education and/or are concerned about their pension are more likely to require care services.*


**Hypothesis** **2.**
*In terms of resource factors, disabled elderly with financial difficulties, lack of family support, and/or willingness to enroll in LCI are more likely to require care services.*


**Hypothesis** **3.**
*In terms of need factors, disabled elderly who have poor health, more chronic diseases and/or greater levels of disability are more likely to need care services.*


## 3. Methods

### 3.1. Sampling

The data is from the 2018 Project of Social Policy Support System for Disadvantaged Families in China, hosted and provided by the project team of “China’s Urban and Rural Family Social Policy Support System Construction”, covering more than 1800 villages in 28 provinces across the country. The survey object is the elderly population aged 60 and above. Samples were screened according to the difficulty of bathing, putting on and taking off clothes, going to the toilet, indoor activities, bowel control, eating, etc. on the Activity of Daily Living (ADL) scale. If there is no difficulty, the elderly person is considered to be fully self-care and excluded from the sample. A total of 6041 data samples were obtained from the survey. After sample screening, a total of 2917 valid samples were retained.

### 3.2. Descriptive Analysis

First, 54.10% of the disabled elderly were male. More disabled elderly lived in urban areas (57.40%) than rural areas (42.60%). The average age of the respondents was 69.52 years. 42.39% of the disabled elderly had completed primary school and 48.50% of them worry about their pension. The majority of the disabled elderly had financial difficulties (70.45%), and only 29.55% of them were in good financial condition. 71.12% of the disabled elderly were able to get help from their families and 60% of disabled elderly were willing to enroll in LCI. Unhealthy (57.57%) and chronic disease (81.80%) were the main characteristics of health. Among the disabled elderly, the slightly disabled elderly accounted for 51.23%, the moderately disabled elderly accounted for 28.26%, and the severely disabled elderly accounted for 20.51%, and the proportion of males was higher than that of females, which was 13.39%, 3.40%, and 1.67%, respectively. In addition, disabled seniors had higher demands for care services in terms of medical services (52.0%), psychological comfort (43.80%) and health education (37.90%) than daily care (31.20%), rehabilitation care (26.30%) and social support (20.30%).

### 3.3. Measures

#### 3.3.1. Explained Variable: Care Service Demand

Based on the literature review, there are six main aspects of the demand for elderly care services, which could be used as the observable variables to composite unobservable latent variables to measure the care service demand of urban and rural disabled elderly [8,20,21,22,23,24]. Care service demand was measured by the item “Your social service needs for the past three months”, and six types of variables were selected to measure the care service demands of the respondents, i.e., urban and rural disabled elderly people, from 6 dimensions of daily care (3 items, meal service, bath service, and housework service), medical treatment (2 items, home medical care and medical escort), health education (1 item, health education service), rehabilitation nursing (1 item, rehabilitation nursing), psychological comfort (2 items, psychological counseling and social work service) and social support (1 item, respite service). In each dimension, if none of the types of services were required, the value was 0, and if any of the types of services were required, the value is 1.

#### 3.3.2. Explanatory Variable

Based on the BMHSU and the theoretical framework, with reference to the results of Dai, Peng, Li, Yang, etc., this article introduces the tendency factors of care service demand with gender, age, education, attributes, and worried pension as explanatory variables, introduces the resource factors of care service demand with the economy, family help and LCI as explanatory variables, and the nursing factors of care service demand with health, chronic disease, and disability degree as explanatory variables. The descriptive statistics of related variables are shown in Appendix A.

### 3.4. Data analysis Methods

#### 3.4.1. Factor Analysis Method

Figure 2 shows the factor analysis method, indicating the care service demands of a disabled elderly can be measured by six observable variables. The demand for care services is a latent variable, which represents the common factor measured by the six observable variables, and $\varepsilon_{1}∼\varepsilon_{6}$ represents the unique variance of each observable variable. EFA and CFA were employed to evaluate the reliability and validity of the latent variables. EFA was used to examine to which extent the items measured constructs care service demand and whether there is a unique common factor for the six indicators. CFA was performed to test whether the overall factor analysis model for measuring care service demand was significant and whether the six indicators were significantly effective for the factor loading coefficient for measuring care service demand, so as to examine whether the factor analysis model for measuring care service demand was valid.

#### 3.4.2. MIMIC Structural Equation Model

The multiple indicators multiple causes model (MIMIC) was employed to explore the influencing factors of the care service demands of the disabled elderly. The MIMIC model in this research implied that gender, age, education, attributes, worried pension, economy, family help, LCI, health, chronic disease, and disability degree would affect the care service of the disabled elderly and the needs of care services were reflected in six aspects: daily care, medical services, health education, rehabilitation care, psychological comfort, and social support. The model is shown in Figure 3.

## 4. Results

### 4.1. Measurement of Care Service Demand

First, the Bartlett test, Kaiser-Meyer-Olkin (KMO) test, and Cronbach’s alpha reliability test were conducted for the six index variables measuring the explained variable “care service demand” ($\chi^{2}\left(15\right)=3301.602$, *p* < 0.001; KMO = 0.823, Cronhach’s α = 0.752). The results indicated that factor analysis could be performed using the six variables of daily care, medical services, health education, rehabilitation care, psychological comfort, and social support.

Second, principal component factor analysis (PCFA), principal factor analysis (PFA), and iterative principal factor analysis (IPFA) were employed to examine if these six variables were measuring a unique common factor. The results are shown in Table 1. PCFA is usually used to do this test, meanwhile, PFA and IPFA are also used to further verify the robustness of the test results. In PCFA, only the eigenvalue of factor 1 is over 1 (2.69), which explains 44.90% of the common variance of these six indicators. PCFA obtain that the factor loadings of the six index variables for factor 1 are all greater than 0.60, indicating that the public factor care service demand has a relatively large correlation coefficient for the six indicators, so it can be accepted and retained. Among the six factors obtained by the PFA, only the eigenvalue of factor 1 is greater than 1 (1.96), and the factor loadings of the six index variables to factor 1 are all greater than 0.50. This shows that the six indicator variables could only measure the only common factor. IPFA also showed that there was only one factor with an eigenvalue greater than 1 (2.14), which explained 83.20% of the common variance of the six indicators, and the factor loadings of each indicator variable to factor 1 were all greater than 0.50, indicating that the six index variables can only measure the unique common factor of care service demand.

Third, CFA was performed to test whether the entire SEM was significant, as shown in Figure 4, and whether the factor loadings of the six indicators were significant. The results explain that the factor model has a high degree of fit ($\chi^{2}\left(6\right)=15.505$, *p* = 0.017, CFI = 0.997, R^2^ = 0.740), and there is no significant difference between the constructed factor model and the real model (RMSEA = 0.023, SRMR = 0.011). The standardized coefficients of the latent variables for the six measurement indicators are all greater than 0.40, and significant at the 0.10% level, and the reliability coefficient ρ = 0.72 for measuring the demands for care services, which exceeds the acceptable standard of 0.70, indicating that the higher the level of care service needs of the investigators, the more likely they are to have demands for daily care, medical services, health education, rehabilitation care, psychological comfort, and social support.

Fourth, to examine the care service demands of disabled elderly in the eastern region, the central region, and the western region, according to Yang & Jia and Li & Yang, six indicators were scored from 0 to 1 [31,49], combined with a bootstrap approach with 1000 replications and CFA (as shown in Figure 4). The results implied that standardized factor loadings corresponding to the obtained six index variables were used as the measurement weight coefficients of each index variable, and the weighted average of the factors was proposed to measure the care service demands of the disabled elderly in urban and rural areas (as shown in Table 2). The results through variance analysis showed that from the overall average of urban and rural areas, care service demands in the eastern region, the central region, and the western region were quite different and the demands for care services in the central and western regions were higher than that in the eastern region, reflecting that the disabled elderly in economically underdeveloped regions had stronger demands for care services. The care services demands of the disabled elderly in rural areas are significantly higher than that of the urban disabled elderly.

### 4.2. Influencing Factor Model Estimation

Based on the MIMIC model, the maximum likelihood method was used to estimate the model, and the model regression results are shown in Figure 5. Combined with the fitting indicators of model (1) in Table 3, the results [CFI = 0.941, RMSEA = 0.037 (*p* < 0.5), SRMR = 0.024 (*p* < 0.5), R^2^(CD) = 0.242] shows that the estimated results of the model are acceptable.

Among the tendency factors, gender, age, attribute, education, and worried pension have a significant impact on the demands for care services of the disabled elderly in urban and rural areas. Hypothesis (1) is verified.

The care service demands of the disabled elderly are affected by gender differences. The regression coefficient of the gender variable is significantly positive at the 0.1% level, indicating that the male elderly have significantly higher demands for nursing services, which is inconsistent with Yin & Du [50]. This may be due to the influence of the research sample of the article. The descriptive statistics showed that the male elderly accounted for 54.1%.

The care service demands of the disabled elderly are affected by age, and the regression coefficient of the age variable is significantly positive at the 5% level, indicating that the nursing service demands of the elderly are significantly higher. The possible reason is that physical function gradually declines with age, which is basically consistent with the research results of Wang & Zheng [15].

There is a difference between urban and rural areas in the demands for care services of the disabled elderly, and the regression coefficient of the urban and rural attribute variables is significantly negative at the 0.1% level, indicating that the urban disabled elderly have significantly lower demands for nursing services, which may be related to the more complete medical technology and transportation facilities and other supporting services in urban areas [51].

The regression coefficient of the education variable is significantly negative at the 0.1% level, indicating that the elderly with higher educational levels have significantly lower demands for nursing services. The possible reason is that the higher the education level, the more likely it is to master health knowledge, reduce the probability of disease occurrence, and have a relatively better self-care ability [52,53,54].

The regression coefficient of the worried pension is significantly positive at the 0.1% level, indicating that the disabled elderly who are worried about “daily care”, “disease care” or “psychological comfort” have a significantly higher demand for care services. The possible reason is that resource allocation is affected by the urban-rural structure, and the disabled elderly have concerns about future care expectations for the elderly, which is consistent with Liu & Guo [55].

Among resource factors, factors such as economic status, help from relatives, and LCI all have an impact on the demands for care services for the disabled elderly in urban and rural areas, which are basically consistent with Hypothesis (2). However, the regression results for the factor “family help” are contrary to the research hypothesis that “disabled elderly people who lack family support are more likely to need care services”.

The regression coefficient of the economic status variable is significantly negative at the 5% level, indicating that the disabled elderly with relatively well-off economic condition have significantly lower demands for nursing services. The possible reason is that the elderly with more affluent economic conditions have stronger purchasing power for services and can obtain high-quality medical services in a timely manner when they are ill. This is consistent with Lei & Wang [40].

The regression coefficient of the family support is significantly positive at the 5% level, indicating that the more family help the disabled elderly seek for, the demands for care services are more significant, which is not consistent with Lei & Li [31,40]. The possible reason, on the one hand, is the sample. The descriptive statistics show that the disabled elderly who can get the help of two or more relatives on a daily basis tend to have an average health status, accounting for 89.1% of the analyzed samples. To a certain extent, it reflects that due to physical dysfunction, the disabled elderly also have a need for more professional care services while receiving help from their relatives in daily life. Especially when the health of the elderly is getting worse and worse, the effect of social care on reducing the time of family care will also weaken, and social care will not completely replace traditional family care [56].On another hand, disabled elderly people who are assisted by their families may also wish to receive respite services from social support, and temporary care services are provided through social sup-port to relieve the pressure and burden of their families’ care. While home care is an alternative to institutional care, it is complementary to community-based home care [57].

The regression coefficient of the LCI variable is significantly positive at the 0.1% level, indicating that the disabled elderly who are willing to participate in LCI have a significantly higher level of demand for nursing services. The possible reason is that on the basis of basic medical insurance and basic old-age insurance, the disabled elderly also expect to be covered by LCI as soon as possible, so that they can obtain services such as medical treatment, nursing, and rehabilitation when there is a need for care. This is consistent with Zhou [26].

Among the need factors, health, chronic disease, and disability degree also significantly affect the demands for care ser-vices of the disabled elderly in urban and rural areas. Hypothesis (3) is verified.

The regression coefficient of the health status variable was significantly negative at the 5% level, indicating that the disabled elderly with good health status had a significantly lower demand for nursing services. The possible reason is that elderly with poorer health expect better quality of life through access to care, which is consistent with Cao & Du [58]. The regression coefficient of the chronic disease status variable was significantly positive at the 10% level, indicating that the disabled elderly with multiple chronic diseases have higher care ser-vices demand. A possible reason is that chronic diseases have a long course of disease, especially the combination of chronic diseases increases the demand for nursing services for the elderly, which is consistent with Kong et al. and Tan et al. [59,60]. The regression coefficient of the disability degree variable is significantly positive at the 0.1% level, indicating that the elderly with more severe disability have significantly higher needs for care services, and the elderly with severe dis-abilities have higher demands for medical services, psychological comfort and life care, consistent with Wu et al. [61].

### 4.3. Model Robustness Examination

The robustness test is carried out using the extended version of the MIMIC model, using the ML method, with the results in Table 3, the 5 model estimation methods for robustness testing were employed: ml + robust; mlmv + robust; bootstrap (bootstrap = 1000); jackknife and gsem.

In the estimation results, part A is a structural model to explore the influencing factors of care service demand. The estimation results of the five robustness tests (2)~(6) are basically consistent with the results of the above maximum likelihood (ML). Part B is the confirmatory factor model (measurement model) for measuring the demand for care services. The results of the five robustness tests are basically consistent with the results in Figure 4 and Figure 5. Part C reports the fit metrics of the MIMIC model. Overall, the path coefficients and their directions shown by the results of the MIMIC model robustness test are consistent with the ML.

### 4.4. Comparison of the Influencing Factors in Different Family Types

According to the results of the CFA (shown in Figure 4), the article further used the factor loading of the six significant variables as the weight coefficients. The score value of each urban and rural disabled elderly’s care service needs was obtained as the explained variable, and the method of multiple linear regression was used to explore and compare the influencing factors of the care service demands of the disabled elderly from urban and rural disadvantaged families and ordinary families. The results are shown in Table 4.

Model (1) is the regression result of the disabled elderly sample from difficult families; Model (2) is the regression result of the disabled elderly sample from ordinary families; Model (3) is the regression result of the overall disabled elderly sample. The overall results are basically consistent with the regression estimation results of the MIMIC structural equation model. It is found that urban and rural attributes, worried pension, and LCI are the common influencing factors of the demand for care services for the disabled elderly in difficult families and ordinary families, and there are also differences between family types.

First, whether from disadvantaged families or from ordinary families, demands for care services are affected by urban and rural attributes, worried pension and LCI, reflecting that the care service needs of the disabled elderly in rural areas cannot be met, and there is a mismatch of supply and demand between the current supply of elderly care services and the growing demand. The disabled elderly is still worried about the future provision of elderly care services. It also indicates the urgency and necessity of implementing the LCI system.

Second, the demands for care services of the disabled elderly in ordinary families are only affected by attributes, worried about pension and LCI, and disabled elderly who live in rural areas, are concerned about pension and/or are willing to enroll in LCI have significantly higher demands for care services, confirming the current uneven distribution of medical care resources between urban and rural areas.

Third, in addition to the influence of attributes, worried about pension and LCI, the care service demands of disabled elderly in disadvantaged families are also related to age, education, family help, health, chronic disease, and disability degree. There are differences in the influencing factors of the demand for care services of the disabled elderly from disadvantaged and ordinary families. In disadvantaged families, older males with lower education, access to families, poorer health, more chronic diseases, and/or greater disability have a significant demand for care services. The possible reason is that majority of the disabled elderly in urban areas participate in the basic old-age insurance system and receive monthly living subsidies. They are more financially secure and can obtain care services easily. For disabled elderly in disadvantaged families, on the one hand, their family’s economic condition is relatively poor, and the purchasing power for care services is limited. On the other hand, the physical health of disabled elderly from disadvantaged families is relatively poor, and most of them have chronic diseases. Although they can get help and care from their families, family members need to work to increase their income, the professionalism and continuity of care services for the disabled elderly are generally poor, and this is consistent with Lin [62].

## 5. Discussion

Data from China’s seventh population census show that aging has entered a stage of rapid development, and the disability of the elderly is an unavoidable social problem. In order to better improve the quality of life and health of the elderly, it is urgent to pay more attention to disability and care. A more sound system of care services should be provided for disabled elderly people in urban and rural areas, especially those from poor families.

Firstly, the elderly care system to meet the needs of urban and rural disabled elderly care services should continue to be improved. The study found that nearly 60% of the disabled elderly in urban and rural areas are willing to participate in nursing insurance, especially the elderly from poor families who show a higher willingness to participate in nursing insurance. Therefore, when gradually expanding the pilot work of the nursing insurance system, we should focus on solving the basic nursing security needs of severely disabled elderly, and pay attention to elderly care assistance. It is necessary to expand the financing channels to improve the fundraising method of social mutual assistance, expand the coverage of the groups and integrate the urban and rural disabled elderly into the coverage of nursing insurance, to provide reasonable cost compensation for the disabled elderly in urban and rural areas to obtain care services, and improve service purchasing ability of the disabled elderly in urban and rural areas.

Secondly, the equalization of basic old-age services between urban and rural areas and between regions should be improved. The research results show that rural disability elderly significantly higher demand for care services, and the central and western disability of the elderly care demand is significantly higher than the eastern region. It’s necessary to optimize care service resource allocation between urban and rural areas as soon as possible, and change the traditional “passive” care services to the “chain” care services that actively discover care demand. On the other hand, broaden financial support for economically underdeveloped rural areas and central and western regions where the demands for care services are higher, expand the supply of elderly care services, and promote the effective connection between the demand for care services for the disabled and the supply of care services.

Thirdly, step up trials of a policy on providing beds for the elderly in families, and explore new models of home care services. The results show that the disabled elderly in urban and rural areas who can get help from their relatives also have a higher demand for care services. With the weakening of family care functions, elderly care is gradually changing from family obligations to public affairs of the whole society [63]. The care problems faced by the disabled elderly cannot be separated from the concerted efforts of family and society. It has become a consensus to strengthen the pilot policy of home care beds and explore a new model of home care services. By integrating formal and informal care resources, providing home care services for the disabled elderly in urban and rural areas, and extending professional care services to families.

Lastly, focus on the needs of care services for disabled elderly people from poor families in urban and rural areas. The results show that the needs for care services of the disabled elderly in difficult families are more significantly affected by propensity factors, resource factors, and need factors, so the disabled elderly in different family types should be given precise policies. The disabled elderly from poor families have a great demand for economic care, and the cost of care, support, and rehabilitation will bring a heavy economic burden to the poor families [64]. On the one hand, efforts should be made to realize a welfare subsidy system covering the disabled elderly from families with economic difficulties, and to achieve an organic connection with the LCI system, improve the payment level of care services for disabled elderly from families with economic difficulties, and reduce the risk of poverty due to illness and return to poverty due to illness. On the other hand, the disabled elderly with economic capacity in ordinary families are encouraged to purchase commercial supplementary old-age care insurance products, so as to improve the security level of care services and reduce the economic burden of care services.

## 6. Study Limitations

The cross-sectional data used in this study cannot explore the dynamic changes in the demand for care services and its influencing factors of the disabled elderly in urban and rural areas from a longitudinal perspective. Additional studies based on panel data are needed to further explore the influencing factors of the demand for care services of the disabled elderly. Since the LCI system is currently in the pilot stage, it mainly covers the insured population of the basic medical insurance for urban employees, and urban and rural residents are temporarily unable to participate in the insurance. Therefore, the lack of evaluation feedback after the use of nursing services by the disabled elderly in urban and rural areas makes the theoretical analysis framework of the BMHSU not fully applied to the research of the article, which needs to be studied after the LCI system has been expanded and fully promoted.

## 7. Conclusions

The research results of this paper verify that the demands for care services of the disabled elderly in urban and rural areas are comprehensively influenced by tendencies, resources, and needs, and also confirm that disability is an important factor for care service demands, so the demands for care services of the disabled elderly in urban and rural areas vary from person to person with different degrees of disability. For the mildly disabled elderly, more attention should be paid to the significant impact of their basic health on the demand for care services. While focusing on improving the quality of life and health, prevention and screening of chronic diseases should be done well. For the elderly with moderate and severe disabilities, more attention should be paid to the mental health of the elderly with disabilities. Through the intervention of effective psychological comfort services, improve the loneliness and depression caused by illness, disability, and the burden of care should be alleviated, and the mental health of the elderly with disabilities.

First, the demands for care services for the disabled elderly in the eastern region (1.09) were significantly lower than that in the central region (1.24) and the western region (1.39). The health of the elderly in the eastern region is generally better, while the proportion of healthy elderly in the central and western regions is relatively lower [65]. Moreover, the long-term care willingness of the elderly in rural areas in different regions is affected by multiple factors, and the economic factor is the most fundamental factor [66], reflecting that the disabled elderly in economically underdeveloped areas are constrained by their physical health and have a stronger demand for care services. Besides, there are still differences between urban and rural areas in the care service demands of the disabled elderly. The care service demand of the disabled elderly in rural areas (1.46) is significantly higher than that of the urban disabled elderly (1.02), reflecting that the unbalanced allocation of medical and nursing resources, the supply of social care services cannot effectively meet the growing demand for care services of the disabled elderly [67].

Second, the demands for care services of the disabled elderly in urban and rural areas are affected by the combination of tendency factors, resource factors, and need factors. Older age, living in rural areas, lower education level, worry about pension, poor financial condition, getting help from families, willing to participate in LCI, poor health, suffering from multiple chronic diseases, and/or higher degree of disability male elderly have significantly higher need for care services. The results not only verify that the demand for care services of the disabled elderly in urban and rural areas is affected by a combination of tendencies, resources, and needs, but also confirm that disability is an important factor in the generation of demand for care services, reflecting the need for urban and rural disabled elderly people to urgent need for care nursing services. In addition, the article also draws a different conclusion from the existing research, the urban and rural disabled elderly who can get help from their relatives have a significantly higher demand for care services. Although the function of home care for the elderly is also gradually weakening, the disabled elderly also expect to receive long-term continuous support from formal care such as home medical care, rehabilitation care, health education, psychological comfort, and access to social support services to provide short respite for family carers. Even though family support may have a surrogate effect on the demands of elderly care services, the findings also reflect that the integration of social care will be an indispensable and important part of family care functions in the future.

Third, the demands for care services of the disabled elderly from ordinary and disadvantaged families are affected by attributes, worried pension, and LCI, but the demands from disadvantaged families are more prominently affected by tendencies, resources and needs, and the demands are greater. Urban and rural attributes, worried pension, and LCI are the common influencing factors for the care service demands of disabled elderly people from ordinary and disadvantaged families, indicating that considering different economic conditions, the care service demands of disabled elderly in rural areas worrying about pension and willing to participate in LCI are significantly higher. This also reflects that the care system to guarantee the disabled elderly people’s equitable access to and utilization of old-age care services is not perfect, especially in rural areas where the disabled elderly people are more eager to get care services. The care service demands for the disabled elderly in ordinary families is only affected by the attributes, worried pension, and LCI, while the demand for care services for the disabled elderly in disadvantaged families is closely related to their gender, age, education level, family help, health, chronic disease, and disability degree. In recent years, although the government has made new explorations and achieved new results in ensuring the basic life of the elderly from poor families, the demands of care services for disabled elderly from poor families have not been effectively met. The possible reasons are the lack of overall design of elderly care services, resulting in inefficient resource allocation of care services in the field of elderly security [68].

## Figures and Tables

**Figure 1 ijerph-19-11112-f001:**
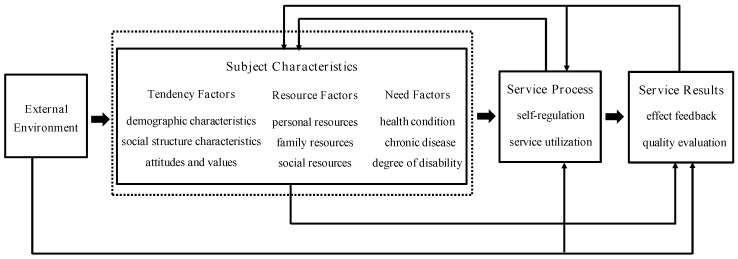
The theoretical analysis framework of the factors affecting the demand for care services for the disabled elderly in urban and rural areas.

**Figure 2 ijerph-19-11112-f002:**
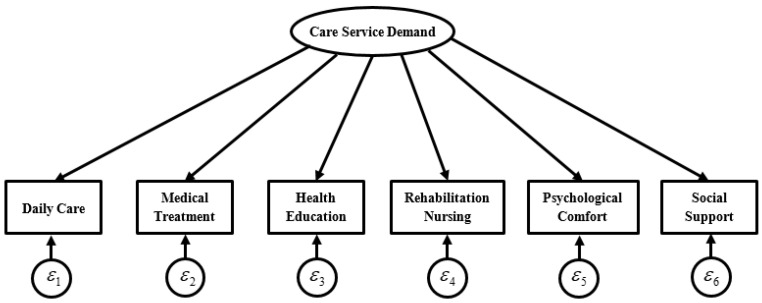
Confirmatory Factor Analysis Model (CFA) for Measuring Care Service Needs.

**Figure 3 ijerph-19-11112-f003:**
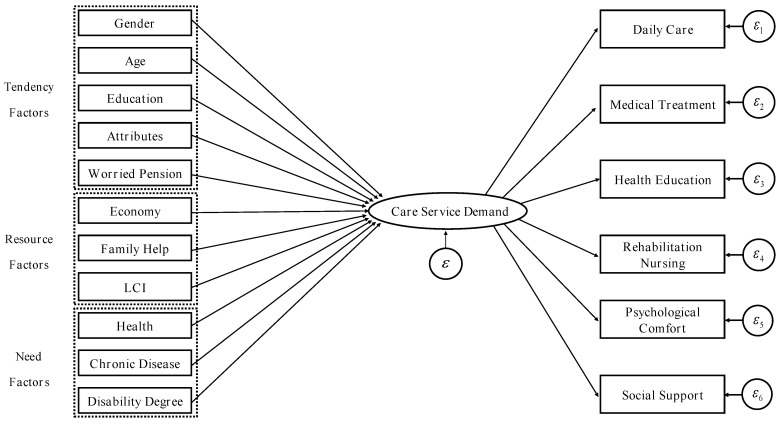
MIMIC model of influencing factors of care service demand.

**Figure 4 ijerph-19-11112-f004:**
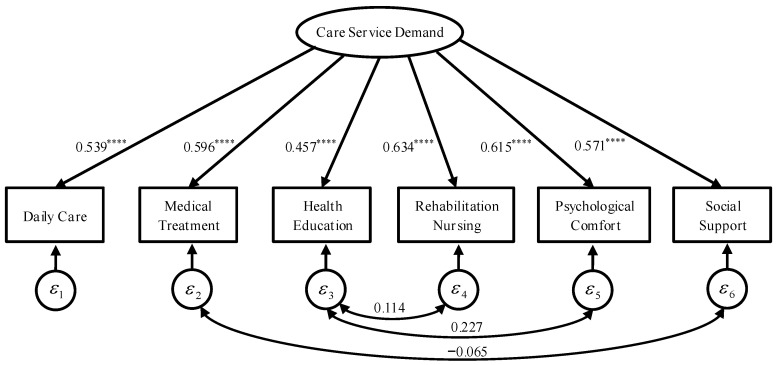
Confirmatory factor analysis of care service needs. Note: the estimation method is resampling method (bootstrap 1000 times), report the standardization coefficient, **** represents the significant level of 0.1%.

**Figure 5 ijerph-19-11112-f005:**
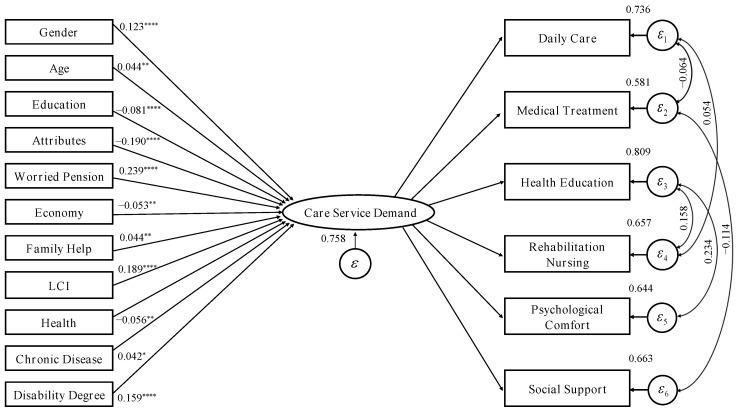
Results of MIMIC model of influencing factors of care service demand. Note: The estimation method is ML, report the standardized coefficient is, ****, **, * are significant at the levels of 0.1%, 5% and 10% respectively.

**Table 1 ijerph-19-11112-t001:** Exploratory Factor Analysis Results (*n* = 2916).

Factor	PCFA		PFA		IPFA	
	Eigenvalue	Factor Loadings	Eigenvalue	Factor Loadings	Eigenvalue	Factor Loadings
Factor 1	2.69	0.62	1.96	0.51	2.14	0.52
Factor 2	0.82	0.66	0.06	0.55	0.25	0.56
Factor 3	0.71	0.65	−0.07	0.55	0.13	0.59
Factor 4	0.67	0.72	−0.07	0.63	0.04	0.67
Factor 5	0.60	0.73	−0.12	0.64	0.01	0.69
Factor 6	0.51	0.64	−0.20	0.53	−0.0002	0.53

Note: Factor loadings is the factor loadings of six index variables in factor 1, Cronhach’s α = 0.746.

**Table 2 ijerph-19-11112-t002:** Calculation of urban and rural care service demand in eastern, central, and western regions.

Weighted Average	Eastern Region	Central Region	Western Region	Variance Analysis
Urban and Rural Weighted Average	1.09	1.24	1.39	F = 17.25 (*p* = 0.00 < 0.05)
Urban Weighted Average	0.88	1.09	1.20	F = 14.70 (*p* = 0.00 < 0.05)
Rural Weighted Average	1.39	1.45	1.61	F = 4.16 (*p* = 0.02 < 0.05)

**Table 3 ijerph-19-11112-t003:** Results of model regression of factors influencing the demand for care services for the disabled.

Model	Estimation Methods	(1)	(2)	(3)	(4)	(5)	(6)
		ml	ml + robust	mlmv + robust	bootstrap	gsem	jackknife
A. Structural Model	Gender	0.06 ****	0.06 ****	0.05 ****	0.06 ****	0.35 ****	0.06 ****
		(5.21)	(5.21)	(5.11)	(5.23)	(5.07)	(5.19)
	Age	0.00 **	0.00 *	0.00	0.00 *	0.01	0.00 *
		(1.99)	(1.94)	(0.76)	(1.92)	(1.46)	(1.93)
	Education	−0.02 ****	−0.02 ****	−0.02 ***	−0.02 ****	−0.10 ***	−0.02 ****
		(−3.43)	(−3.51)	(−3.24)	(−3.67)	(−2.65)	(−3.49)
	Attributes	−0.09 ****	−0.09 ****	−0.10 ****	−0.09 ****	−0.54 ****	−0.09 ****
		(−7.91)	(−7.60)	(−8.85)	(−7.45)	(−7.40)	(−7.57)
	Worried Pension	0.11 ****	0.11 ****	0.06 ****	0.11 ****	0.72 ****	0.11 ****
		(9.64)	(9.51)	(6.27)	(9.27)	(9.04)	(9.47)
	Econmy	−0.01 **	−0.01 **	−0.02 **	−0.01 **	−0.08 *	−0.01 **
		(−2.23)	(−2.24)	(−2.49)	(−2.23)	(−1.92)	(−2.23)
	Family Help	0.01 **	0.01 **	0.01 *	0.01 **	0.05 **	0.01 **
		(2.00)	(2.00)	(1.70)	(2.04)	(2.54)	(1.99)
	LCI	0.09 ****	0.09 ****	0.10 ****	0.09 ****	0.59 ****	0.09 ****
		(8.07)	(8.33)	(8.57)	(8.08)	(7.94)	(8.30)
	Health	−0.01 **	−0.01 **	−0.02 ***	−0.01 **	−0.06	−0.01 **
		(−2.24)	(−2.23)	(−2.84)	(−2.24)	(−1.55)	(−2.22)
	Chronic Disease	0.01 *	0.01 *	0.01 **	0.01 *	0.05 **	0.01 *
		(1.79)	(1.77)	(2.47)	(1.67)	(2.39)	(1.76)
	Disability Degree	0.05 ****	0.05 ****	0.07 ****	0.05 ****	0.26 ****	0.05 ****
		(6.48)	(5.87)	(8.87)	(5.95)	(5.47)	(5.84)
B. Confirmatory Factor Model (Measurement Model)	Daily Care	1.00	1.00	1.00	1.00	1.00	1.00
	_cons	0.07	0.07	0.16 ***	0.07	—	0.07
		(1.09)	(1.08)	(2.74)	(1.06)	—	(1.08)
	Medical Treatment	1.42 ****	1.42 ****	1.32 ****	1.42 ****	1.32 ****	1.42 ****
		(17.17)	(17.31)	(19.93)	(17.75)	(11.29)	(17.32)
	_cons	0.19 **	0.19 **	0.32 ****	0.19 **	—	0.19 **
		(2.32)	(2.31)	(4.21)	(2.26)	—	(2.30)
	Health Education	0.93 ****	0.93 ****	0.87 ****	0.93 ****	1.09 ****	0.93 ****
		(13.67)	(12.88)	(14.66)	(12.41)	(11.14)	(12.89)
	_cons	0.18 ***	0.18 ***	0.25 ****	0.18 ***	—	0.18 ***
		(3.27)	(3.24)	(4.83)	(3.18)	—	(3.23)
	Rehabilitation Nursing	1.08 ****	1.08 ****	1.08 ****	1.08 ****	1.67 ****	1.08 ****
		(17.69)	(16.73)	(20.18)	(16.70)	(10.98)	(16.76)
	_cons	0.01	0.01	0.10	0.01	—	0.01
		(0.10)	(0.10)	(1.58)	(0.10)	—	(0.10)
	Psychological Comfort	1.29 ****	1.29 ****	1.21 ****	1.29 ****	1.59 ****	1.29 ****
		(16.69)	(16.17)	(18.49)	(16.29)	(10.97)	(16.18)
	_cons	0.16 **	0.16 **	0.25 ****	0.16 **	—	0.16 **
		(2.05)	(2.03)	(3.61)	(1.99)	—	(2.02)
	Social Support	1.01 ****	1.01 ****	0.94 ****	1.01 ****	1.33 ****	1.01 ****
		(16.10)	(15.32)	(16.78)	(15.42)	(11.16)	(15.35)
	_cons	−0.01	−0.01	0.06	−0.01	—	−0.01
		(−0.18)	(−0.18)	(1.13)	(−0.17)	—	(−0.18)
C. Fit Metrics	*n*	2362	2362	2917	2362	2365	2362
	R^2^ (CD)	0.242	0.242	0.228	0.242	—	0.242
	CFI	0.941	—	—	0.941	—	0.941
	RMSEA	0.037	—	—	0.037	—	0.037
	SRMR	0.024	0.024	—	0.024	—	0.024

Note: report denormalization factor, (3) is the result of bootstrap 1000 times, ****, ***, **, * are significant at the levels of 0.1%, 1%, 5% and 10% respectively, “—” means the value is blank.

**Table 4 ijerph-19-11112-t004:** Comparison of the results of factors influencing the needs of elderly care services in difficult families and ordinary families.

Model	(1)	(2)	(3)
	Difficult Families	Ordinary Families	Overall Families
Gender	0.256 ****	0.015	0.216 ****
	(5.53)	(0.16)	(5.23)
Age	0.005 *	0.001	0.005 *
	(1.67)	(0.10)	(1.69)
Education	−0.076 ***	−0.015	−0.068 ***
	(−2.98)	(−0.34)	(−3.09)
Attributes	−0.357 ****	−0.284 ***	−0.332 ****
	(−7.58)	(−3.03)	(−7.93)
Worried Pension	0.442 ****	0.497 ****	0.452 ****
	(9.66)	(5.68)	(11.14)
Family Help	0.031 **	0.027	0.027 **
	(2.06)	(0.96)	(2.09)
LCI	0.352 ****	0.394 ****	0.363 ****
	(7.73)	(4.40)	(8.95)
Health	−0.045 *	−0.081	−0.052 **
	(−1.71)	(−1.54)	(−2.23)
Chronic Disease	0.027 **	0.016	0.026 **
	(2.02)	(0.56)	(2.13)
Disability Degree	0.200 ****	0.065	0.185 ****
	(6.54)	(0.90)	(6.62)
_cons	0.167	0.648	0.213
	(0.66)	(1.19)	(0.93)
*n*	1935	427	2362

Note: ****, ***, **, * are significant at the levels of 0.1%, 1%, 5% and 10% respectively.

## Data Availability

Data sharing not applicable. The data are not publicly available due to they are derived from official surveys conducted by the Ministry of Civil Affairs of the People’s Republic of China.

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
