# Peer review of "Study on the Measurement and Influencing Factors of Care Service Demand of Disabled Elderly in Urban and Rural China"

_ijerph, 2022, doi:10.3390/ijerph191711112_

Round 1
Reviewer 1 Report
In the era of the growing elderly population around the world, this study is particularly important to understand the influencing factors behind the unmet care demands of the disabled elderly. This paper provides findings underpinned by strong statistical analysis.
The introduction needs to be strengthened with a stronger justification for this research.
Author Response
Response to Reviewer 1 Comments
Manuscript Number:ijerph-1833328
Manuscript Title: Study on the Measurement And Influencing Factors of Care Service Demand of Disabled Elderly in Urban And Rural China
Point 1: In the era of the growing elderly population around the world, this study is particularly important to understand the influencing factors behind the unmet care demands of the disabled elderly. This paper provides findings underpinned by strong statistical analysis.
Response 1:
Thanks for your comments.
We paid special attention to the revision according to your comments, and please review the revised manuscript for details.
Moreover, we will reply to the comments point as follows, and all the responses are in red to make it easier to be noticed.
Point 2:The introduction needs to be strengthened with a stronger justification for this research.
Response 2:
Thanks for your comments.
We are sorry that we did not provide strong justification for this research in the introduction. In this revised manuscript, we have added the following contents in introduction as you suggested.
In the introduction, we have added background on the demands of care services for the elderly, focusing on the definition of care services proposed by the World Health Organization and the World Bank, and the interpretation of care services proposed by Chinese scholars.
At the same time, combined with the current pilot of China's long-term care insurance system and the implementation of the care service policy for the disabled elderly, we pointed out the problems related to the demands of care services for the disabled elderly, and provided a policy basis for the study.
In addition, according to the existing research literature, the manuscript points out that there are few research on the demands of care services for the disabled elderly. The existing literature lacks a systematic theoretical analysis framework and does not have a deep enough understanding of the influencing factors affecting the demand for elderly care services, which cannot comprehensively reflect the demand for elderly care services.
Please refer to the revised manuscript for more details.
Reference:
World Health Organization. Establishing an international consensus on long-term care policies for the elderly. World Health Organization, 2000.
Tang, J.; Feng, L. A Global Consensus and Concept Frame for Long-term Care. Social Policy Research 2021, 1,18-38. (In Chinese)
Dai, W. Elderly demands for long-term care and its influencing factors—based on the survey of Anhui and Jiangsu. Popu-lation Research 2011, 4, 86-94. (In Chinese)
Wang, X.; Zheng, C. Empirical analysis of the elderly health and long-term care. Journal of Shandong University (Phi-losophy and Social Sciences) 2014, 3, 30-41. (In Chinese)
Zeng, W.; Hu, J.; Zhang, R.; Wu, Y. Social determinants of long-term care needs for rural needy elderly in China—taking Ankang as an example. Journal of Xi'an Jiaotong University (Social Science) 2014, 4, 61-68. (In Chinese)
Tang, J.; Feng, L. A global consensus and conceptual framework for long-term care. Social Policy Research 2021, 1, 18-38. (In Chinese)
Cao, Y.; Chen, J.; Lian, H.; Liu, P. The influencing factors of long-term care insurance demand in China: An empir-ical study based on Jiangsu Province. Chinese Journal of Health Policy 2018, 4, 19-23. (In Chinese)
Liao, X. The demand for long-term care services and its influencing factors in rural disabled elderly—An empirical evidence based on province Hunan. Population and Development 2019, 1, 119-128. (In Chinese)

Reviewer 2 Report
This article concerns caregiving services of the disabled elderly from the perspective of needs and influencing factors, combining the BMHSU and survey data. In summary, this article is interesting and well structured, with findings presented in an easily understandable way. And I believe the methodology taken is appropriate for this type of study. However, according to some minor errors, corrections needed to be made by referring to the comments below.
In the section Introduction, 1) more materials about care service, such as the definition in the policies, practical examples or others, should be added between paragraphs 2 and 3.
In the section on Literature review and theory, 1) “ointed” in the “Zhu & Guo ointed out” in page 4 should be corrected to “pointed”. 2) The contents about “External Environment”, “Service Process” and “Service Results” in Figure 1 seem not to be shown much in the article. How to consider putting it in the framework?
In the section of Methods,1) What is the source of the six types of variables in 3.3.1? 2)Where is the Appendix 1? 3) ε1 to ε6 represent different variables in Figure 2 and Figure 3. What’s the consideration? 4) In Figure 3, care service demand and other six variables own the same symbols, are they parallel?
In the section of Results, 1) What do the results obtained by PFA indicate us in 4.1? 2) “as showed in Figure 4” on page 10 under Table 1 should be corrected to “shown”. 3) What is the meaning of the values between the variables such as 0.114, 0.227, and -0.065 in Figure 4 and the values in the right of Figure 5? 4) A blank should be added in the “thatthe” in 4.2 above the Figure 5. 5) Is the “care concept” similar to “worried pension”?
Overall, the paper requires major English reviews.
Author Response
Response to Reviewer 2 Comments
Manuscript Number:ijerph-1833328
Manuscript Title: Study on the Measurement And Influencing Factors of Care Service Demand of Disabled Elderly in Urban And Rural China
Point 1:This article concerns caregiving services of the disabled elderly from the perspective of needs and influencing factors, combining the BMHSU and survey data. In summary, this article is interesting and well structured, with findings presented in an easily understandable way. And I believe the methodology taken is appropriate for this type of study. However, according to some minor errors, corrections needed to be made by referring to the comments below.
Response 1:
Thank you for your affirmation and praise for our work.
We would like to express our sincere gratitude and appreciation to your valuable suggestions and positive comments that will help to improve the quality of the manuscript. A point by point response is included below. Moreover, we will reply to the comments point as follows, and all the responses are in red to make it easier to be noticed.
Point 2: In the section Introduction, 1) more materials about care service, such as the definition in the policies, practical examples or others, should be added between paragraphs 2 and 3.
Response 2:
Thank you for the advice.
In this revised manuscript, we have added the following in introduction as you suggested.
In the introduction, we have added background on the demands of care services for the elderly, focusing on the definition of care services proposed by the World Health Organization and the World Bank, and the interpretation of care services proposed by Chinese scholars.
At the same time, combined with the current pilot of China's long-term care insurance system and the implementation of the care service policy for the disabled elderly, we pointed out the problems related to the demands of care services for the disabled elderly, and provided a policy basis for the study.
In addition, according to the existing research literature, the manuscript points out that there are few research on the demands of care services for the disabled elderly. The existing literature lacks a systematic theoretical analysis framework and does not have a deep enough understanding of the influencing factors affecting the demand for elderly care services, which cannot comprehensively reflect the demand for elderly care services.
Please refer to the revised manuscript for more details.
Reference:
World Health Organization. Establishing an international consensus on long-term care policies for the elderly. World Health Organization, 2000.
Tang, J.; Feng, L. A Global Consensus and Concept Frame for Long-term Care. Social Policy Research 2021, 1,18-38. (In Chinese)
Dai, W. Elderly demands for long-term care and its influencing factors—based on the survey of Anhui and Jiangsu. Popu-lation Research 2011, 4, 86-94. (In Chinese)
Wang, X.; Zheng, C. Empirical analysis of the elderly health and long-term care. Journal of Shandong University (Phi-losophy and Social Sciences) 2014, 3, 30-41. (In Chinese)
Zeng, W.; Hu, J.; Zhang, R.; Wu, Y. Social determinants of long-term care needs for rural needy elderly in China—taking Ankang as an example. Journal of Xi'an Jiaotong University (Social Science) 2014, 4, 61-68. (In Chinese)
Tang, J.; Feng, L. A global consensus and conceptual framework for long-term care. Social Policy Research 2021, 1, 18-38. (In Chinese)
Cao, Y.; Chen, J.; Lian, H.; Liu, P. The influencing factors of long-term care insurance demand in China: An empir-ical study based on Jiangsu Province. Chinese Journal of Health Policy 2018, 4, 19-23. (In Chinese)
Liao, X. The demand for long-term care services and its influencing factors in rural disabled elderly—An empirical evidence based on province Hunan. Population and Development 2019, 1, 119-128. (In Chinese)
Point 3: In the section on Literature review and theory, 1) “ointed” in the “Zhu & Guo ointed out” in page 4 should be corrected to “pointed”.
Response 3:
We are very sorry that our negligence leads to inaccurate expression, and we have corrected the original text as you suggested.
Point 4: In the section on Literature review and theory, 2) The contents about “External Environment”, “Service Process” and “Service Results” in Figure 1 seem not to be shown much in the article. How to consider putting it in the framework?
Response 4:
Thanks for your comments.
First of all, referring to previous scholars' research on Anderson's Behavioral Model of Health Services Use, in this study, we also provide a relatively complete theoretical analysis framework to better present the integrity of the theoretical framework. Based on this comprehensive theoretical analysis framework, this manuscript could have a more integrated analysis perspective while focusing on the core content, which is in the dotted box.
Secondly, although we do not incorporate the external environment, service process and service results into quantitative and empirical analysis, our study also mentioned and the “External Environment”, “Service Process” and “Service Results” in introduction and disscussion in this manuscript, in explicit and implicit ways. In the introduction, we describe and analyze the establishment of care service system and the pilots of long-term care insurance, which provides good policy guidance for our research. In addition, in the analysis process and final discussion section, we also carry out potential analysis and discussion around the “Service Process” and “Service Results”.
Finally, as the key part of the theoretical analysis framework, if the three parts (“External Environment”, “Service Process” and “Service Results”) were removed, only focusing on the key part (“Subject characteristics”), this analysis would lack the overall environment and macro nature, leading a series of limitations of lack of integrity and completeness.
Therefore, we hope that you could agree we continue to maintain this theoretical analysis framework in this manuscript.
Point 5: In the section of Methods,1) What is the source of the six types of variables in 3.3.1? 2)Where is the Appendix 1?
Response 5:
Thanks for your comments.
Based on the literature review, there are six main aspects of the demend for elderly care services, which could be used as the observable variables to composite unob-servable latent variable to measure the care service demand of urban and rural disabled elderly.
We are very sorry that you didn't see the appendix 1 (Supplementary Materials) we uploaded. Based on the existing literature on the measurement of the demand for care services for disabled elderly, especially the definition of care services in the World Health Organization's International Consensus on Building Long-term Care Policies for Older Persons and the Global Report on Ageing and Health, as well as the research of other scholars (Chen W., 2002; Pei X. & Fang J., 2010; Zhang Y., 2015; Dai W., 2018) in the field, it is found that researchers generally reached a consensus on the content of care services that six ascepts (daily care, medical treatment, health education, rehabilitation nursing, psychological comfort and social support) usually be used to measure the demand for care services for urban and rural disabled elderly. We have added the information into revised manuscript. Meanwhile, you could find the information on the six ascepts in “Supplementary Materials” at the end of this revised manuscript . (At the end of the manuscript, we add a part of the appendix, mainly for the research involved in the explanatory variables and the explanatory variables descriptive statistical analysis were supplemented.)
Please refer to the revised manuscript for details.
Reference:
World Health Organization. Establishing an international consensus on long-term care policies for the elderly. World Health Organization, 2000.
World Health Organization. Global report on aging and health. World Health Organization, 2015.
Chen, W. The reforms in tendance service system for senior citizen in developed countries and its use for reference. Nankai Journal 2002, 3,58-64. (In Chinese)
Pei, X.; Fang, L. Introduction to long-term care for the Elderly. Beijing: Social Science Literature Publishing House, 2010. (In Chinese)
Zhang, Y. long-term care for the elderly: Institutional selection and international comparison. Beijing: Economics and Management Press, 2015. (In Chinese)
Dai, W. Research on the Construction of China's long-term care Service System. Beijing: Social Science Literature Pub-lishing House, 2018. (In Chinese)
Point 6: In the section of Methods, 3) ε1 to ε6 represent different variables in Figure 2 and Figure 3. What’s the consideration?
Response 6:
Thanks for your valuable suggestion.
We are very sorry that because of our carelessness, the error variables ε1 ~ε6 were mismarked in Figure 3. ε1 ~ε6 in Figure 3 indicate the measurement error items of six observable variables, and should be consistent that in Figure 2. We have revised Figure 3 in this revised manuscript.
Please refer to the revised manuscript for details.
Point 7: In the section of Methods, 4) In Figure 3, care service demand and other six variables own the same symbols, are they parallel?
Response 7:
Thank you so much for your helpful suggestion.
There is no parallel relationship between care service demand and other six variables (daily care, medical treatment, health education, rehabilitation nursing, psychological comfort, social support), and six variables are used as observable variables to synthesize the latent variable of care service demand. The results of exploratory factor analysis and confirmatory factor analysis show that the demands of urban and rural disabled elderly care services can be measured by daily care, medical treatment, health education, rehabilitation care, psychological comfort and social support.
The arrows corresponding to the care service demand and the six significant variables are marked with the load coefficient, indicating that for every 1 unit increase in the standard deviation of the latent variable (care service demand), the standard deviation of the six care services will increase the load coefficient corresponding to the arrows.
In addtion, we are very sorry for the carelessness to mismark the measurement error items (ε1~ε7) in Figure 3 in the previous manuscript, and we have made changes (ε, ε1~ε6), and please refer to the figure 3 in this revised manuscript.
Point 8: In the section of Results, 1) What do the results obtained by PFA indicate us in 4.1?
Response 8:
Thanks for your comments.
Generally, in the process of exploratory factor analysis, the academia uses PCFA, PFA, IPFA and other methods to conduct interactive tests on the analysis results. In this research, the interaction test of PCFA, PFA and IPFA is also used. We added a supplementary explanation in this revised manuscript.
From the results, only one factor with eigenvalue greater than 1 can be obtained by PCFA, and the loadings of factor 1 to the six indicator variables are greater than 0.5, which proves that one latent variable could be extracted from the six observable variables, that is, care service demand could be used to represent the six observable variables. Subsequent analysis results of PFA and IPFA further verified the analysis results of PCFA.
Therefore, the analysis results of PFA show that a latent variable care service demand could be synthesized through six observable variables, which proves that the analysis results of PCFA are reliable.
Thanks for your valuable suggestion again.
Point 9: In the section of Results, 2) “as showed in Figure 4” on page 10 under Table 1 should be corrected to “shown”.
Response 9:
Thanks for your comments.
Due to our carelessness, resulting in inaccurate grammar use and expression, we have revised in this revised manuscript .
Point 10: In the section of Results, 3) What is the meaning of the values between the variables such as 0.114, 0.227, and -0.065 in Figure 4 and the values in the right of Figure 5?
Response 10:
Thanks for your comments.
The values between the variables such as 0.114, 0.227 and -0.065 in Figure 4 and the values in the right of Figure 5 are the results of the adjustment of the structural equation model.
In the structural equation model in this study, we set the error terms of the variable indexes are correlated in order to correct the potential covariance between variables so as to improve the fitting accuracy of the model eventually. The analysis was conducted with Stata 15.0, and the command “estate mindices” was used in the option.
Figure 4 and the values in the right of Figure 5 (such as 0.114, 0.227 and-0.065) represent the correlation coefficient between the two error items, that is the ratio of the covariance of the two variables to their standard deviations.
It is regrettable that due to the limited length of the manuscript, the detailed explanation of the analysis results could not be included in this manuscript.
Please refer to the revised manuscript for details.
Point 11: In the section of Results, 4) A blank should be added in the “thatthe” in 4.2 above the Figure 5.
Response 11:
Thanks for your comments.
We apologize for the mistake. We have checked and corrected the mistake.
Point 12: In the section of Results, 5) Is the “care concept” similar to “worried pension”?
Response 12:
Thanks for your comments.
It is a pity that, due to our negligence, the “care concept” in “4.2. influence factor model estimation” is a mistake, and the correct content should be “worried pension”. We have corrected in this resived manuscript.

Round 2
Reviewer 2 Report
The authors addressed my comments appropriately and significantly improved the paper. The paper can be accepted in the present form.